# GLGait: A Global-Local Temporal Receptive Field Network for Gait Recognition in the Wild

Guozhen Peng
State Key Laboratory of Virtual
Reality Technology and Systems,
Beihang University
guozhen_peng@buaa.edu.cn

Yunhong Wang
State Key Laboratory of Virtual
Reality Technology and Systems,
Beihang University
yhwang@buaa.edu.cn

Yuwei Zhao
State Key Laboratory of Virtual
Reality Technology and Systems,
Beihang University
sy2206328@buaa.edu.cn

Shaoxiong Zhang
The School of Communication
Engineering, Hangzhou Dianzi
University
zhangsx@hdu.edu.cn

Annan Li*
State Key Laboratory of Virtual
Reality Technology and Systems,
Beihang University
liannan@buaa.edu.cn

## Abstract

Gait recognition has attracted increasing attention from academia and industry as a human recognition technology from a distance in non-intrusive ways without requiring cooperation. Although advanced methods have achieved impressive success in lab scenarios, most of them perform poorly in the wild. Recently, some Convolution Neural Networks (ConvNets) based methods have been proposed to address the issue of gait recognition in the wild. However, the temporal receptive field obtained by convolution operations is limited for long gait sequences. If directly replacing convolution blocks with visual transformer blocks, the model may not enhance a local temporal receptive field, which is important for covering a complete gait cycle. To address this issue, we design a Global-Local Temporal Receptive Field Network (GLGait). GLGait employs a Global-Local Temporal Module (GLTM) to establish a global-local temporal receptive field, which mainly consists of a Pseudo Global Temporal Self-Attention (PGTA) and a temporal convolution operation. Specifically, PGTA is used to obtain a pseudo global temporal receptive field with less memory and computation complexity compared with a multi-head self-attention (MHSA). The temporal convolution operation is used to enhance the local temporal receptive field. Besides, it can also aggregate pseudo global temporal receptive field to a true holistic temporal receptive field. Furthermore, we also propose a Center-Augmented Triplet Loss (CTL) in GLGait to reduce the intra-class distance and expand the positive samples in the training stage. Extensive experiments show that our method obtains state-of-the-art results on in-the-wild datasets, *i.e.*, Gait3D and GREW. The code is available at https://github.com/bgdpgz/GLGait.

*Corresponding author.

## CCS Concepts

• **Computing methodologies → Biometrics**.

## Keywords

Gait Recognition, In the Wild, Global-Local Temporal Receptive Field, Gait Silhouette Sequence, Neural Network

**ACM Reference Format:**
Guozhen Peng, Yunhong Wang, Yuwei Zhao, Shaoxiong Zhang, and Annan Li. 2024. GLGait: A Global-Local Temporal Receptive Field Network for Gait Recognition in the Wild. In *Proceedings of the 32nd ACM International Conference on Multimedia (MM '24), October 28–November 1, 2024, Melbourne, VIC, Australia.* ACM, New York, NY, USA, 10 pages. https://doi.org/10.1145/3664647.3680812

## 1 Introduction

Gait recognition is a technique that identifies pedestrians by analyzing their walking patterns. Unlike other biometric characteristics such as the face and iris, gait can be captured from a distance without pedestrian cooperation, thus attracting increasing attention.

Many advanced appearance-based methods [4, 12, 20, 28] using silhouette sequences as input have obtained successful performance on in-the-lab datasets such as CASIA-B [51] and OU-MVLP [40]. However, the performance of these methods drops dramatically on in-the-wild datasets, *i.e.*, Gait3D [52] and GREW [54]. The primary reason lies in that the scenario of existing in-the-lab datasets differs greatly from in-the-wild ones. In wild scenarios, pedestrians may walk at varying velocities or follow a non-straight routine, and even can be occluded by other pedestrians or objects. These noisy factors are challenging for methods designed according to in-the-lab datasets, resulting in performance degradation.

To address the aforementioned issue, we first compare silhouette sequences between CASIA-B [51] and Gait3D [52] datasets. As shown in Figure 1 (a), multiple and evenly distributed gait circles can be clearly observed from in-the-lab sequences (see pedestrian #1). Thus, appropriate local temporal receptive field can facilitate the model in learning the pattern of a complete gait circle. While in wild scenarios, (see pedestrian #2) such ideal temporal segments are no longer available, since the variations of pace and walking directions have a great influence on the appearance. Therefore, a global temporal receptive field is necessary for aligning gait patterns.

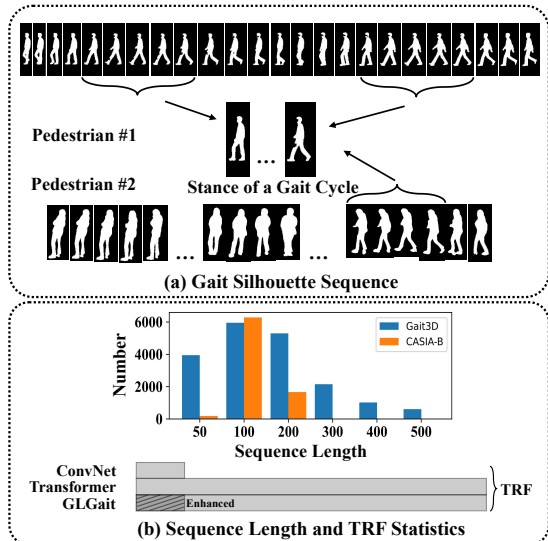

**Figure 1: Comparison of local and global temporal receptive field. (a) Gait cycles are evenly distributed in laboratory scenarios (Pedestrian #1), thus proper-sized local receptive field can capture a complete cycle. While in the wild (Pedestrian #2) the distribution is sparse and random, which implies a larger receptive field is necessary. Corresponding sequences are sampled from CASIA-B [51] and Gait3D [52], respectively. (b) Sequence length and TRF statistics in CASIA-B and Gait3D, where TRF is the temporal receptive field.**

Recently, some methods [10, 11, 46] have been proposed to address the issue of gait recognition in the wild. These methods are mainly centered around Convolutional Neural Networks (ConvNets). Compared with methods designed according to in-the-lab datasets, they often incorporate more convolutional operations to exhibit a larger receptive field. Given that gait silhouettes are typically quite small, such as $64 \times 64$ pixels, the spatial receptive field of these methods is already sufficient. However, the temporal receptive field obtained through convolutional operations is significantly insufficient. As shown in Figure 1 (b), the temporal receptive field usually can only cover around 50 silhouettes in a sequence, which is quite limited in contrast to typical frame number of in-the-wild gait sequences. The limited local temporal receptive field struggles to capture adequate information about pedestrian body shape changes, thus a global temporal receptive field is essential.

Some works [2] use visual transformer block [7] to obtain a global temporal receptive field. However, directly replacing convolution blocks with transformer blocks cannot necessarily enhance a local temporal receptive field. Considering that a transformer block usually applies multi-head self-attention [44] (MHSA) to obtain a global receptive field, adding MHSA before temporal convolution operation in ConvNets is a possible resolution to obtain a global-local temporal receptive field. However, due to using the output of ConvNets as input to MHSA, dimension explosion occurs in token size with large channels in ConvNets, such as 512, resulting in high memory consumption and computation cost. To address the issue, we propose Pseudo Global Temporal Self-Attention (PGTA). Compared with MHSA, PGTA reduces complexity in two aspects. First,

considering the receptive field issue in the temporal dimension, we separate the spatial dimension, only calculating the patch size from the temporal dimension. Secondly, we separate the patch size from tokens in PGTA inspired by [33], obtaining a pseudo temporal receptive field of each element in tokens. With the same temporal convolution kernel size and patch size, these pseudo global temporal receptive fields are naturally aggregated to a truly global one. We name the combination of PGTA and temporal convolution operation as Global-Local Temporal Module (GLTM).

Based on GLTM, we design a Global-Local Temporal Receptive Field Network, named GLGait. The backbone of GLGait consists of a vision encoder and Global-Local 3D (GL-3D) Blocks. In the vision encoder, GLGait encodes a preliminary pedestrian representation. For GL-3D block, in the temporal dimension, GLTM is used to effectively obtain a global-local temporal receptive field. While in the spatial dimension, we use 2D convolution operations, the reason is straightforward: the spatial receptive field is already sufficient by convolution operations.

Furthermore, inspired by center loss [17, 48], we also propose a simple yet effective loss function, named Center-Augmented Triplet Loss (CTL) to assist in model training as a component in GLGait. Based on conventional triplet loss [18], CTL additionally considers the class center as a positive sample for each input. This operation has two advantages: reducing the intra-class distance and expanding the positive samples in the training stage.

The main contributions are summarized as follows:

1) We design a Global-Local Temporal Receptive Field Network (GLGait) to obtain a global-local temporal receptive field for gait recognition in the wild.

2) We propose Pseudo Global Temporal Self-Attention (PGTA) to reduce the high memory and computation complexity of multi-head self-attention [44] (MHSA).

3) We propose a Center-Augmented Triplet Loss (CTL) to assist in model training. CTL can reduce the intra-class distance and expand the positive samples in the training stage.

4) Extensive experiments demonstrate that our approach obtains the state-of-the-art performance on in-the-wild datasets, *i.e.*, Gait3D [52] and GREW [54].

## 2 Related Works

### 2.1 Model-based Gait Recognition

Model-based methods [1, 13, 25–27, 35, 41, 42, 49] consider the physical structure of the human body and utilize pose information as input, such as 2D skeletons, 3D joints, and point clouds. For instance, PoseGait [27] utilizes the human body pose information to extract temporal-spatial features and employs ConvNets to extract high-level temporal-spatial features. GaitGraph [42] proposes a pose estimator to extract pose features and adopts graph convolutional neural networks [39] for gait recognition. GPGait [13] uses a unified pose representation as input. Then a part-aware graph convolutional network is proposed to enable efficient graph partition and local-global spatial feature extraction. LidarGait [38] leverages LiDAR to generate gait point clouds for gait recognition. Model-based methods are robust in some scenarios, such as clothes changed [5, 37]. However, pose information is not easy to calculate, thus these methods may be difficult to apply in a new scenario.

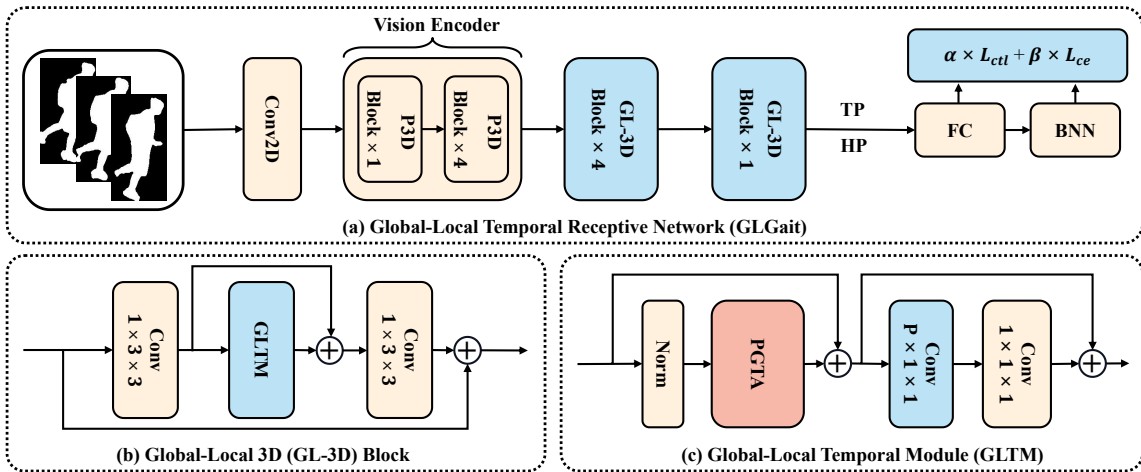

**Figure 2: Pipeline of the proposed GLGait. The backbone mainly consists of the vision encoder and GL-3D blocks. Specifically, we use Pseudo Global Temporal Self-Attention (PGTA) to extract global temporal information and a temporal convolution operation to enhance the local temporal information extraction in Global-Local Temporal Module (GLTM). TP denotes the Temporal Max Pooling operation, HP is the Horizontal Pooling operation [11, 14], FC is the separate fully connected layers [4], and BNN is BNNeck [32]. The final loss function is composed of a center-augmented triplet loss (CTL) and a cross-entropy loss.**

## 2.2 Appearance-based Gait Recognition

Appearance-based methods are employed to learn the body appearance without achieving explicit structures. Most methods [3, 4, 8–10, 12, 19, 23, 28, 45, 46, 50] use silhouette sequences as inputs and use deep neural networks to extract feature [16, 22, 36, 43]. Then, by comparing similarities with other silhouette sequences, the recognition is implemented. Some methods [47, 53] also utilize semantic parsing of pedestrians as the input, obtaining satisfactory performance.

Previous works [4, 6, 12, 28] focus on in-the-lab scenarios, obtaining successful performance. However, with the recent development of in-the-wild datasets, *i.e.*, Gait3D [52] and GREW [54], these methods are facing new challenges. To employ precise gait recognition in wild scenarios, some methods are proposed. GaitGCI [9] utilizes counterfactual intervention learning to eliminate the impact of confounder, focusing on the discriminative and interpretable regions effectively. DyGait [46] focuses on the extraction of dynamic features and proposes a dynamic augmentation module to learn part features automatically. Fan et al. [11] proposes a simple yet effective ResNet-like [16] framework, which is referred as GaitBase. Based on GaitBase, DGaitV2 [10] obtains higher accuracy by increasing the number of stacked blocks.

Compared to pose information, appearance cues are relatively easy to obtain in a new scenario, exhibiting broader adaptability. However, appearance information is susceptible to the influence of pedestrian appearance, exhibiting low robustness in certain specific scenarios such as clothes change [5, 37]. Our GLGait belongs to the appearance-based category, using gait silhouette sequences as input and establishing a global-local temporal receptive field.

## 2.3 Gait Transformers

Vision transformer [7] has achieved successful performance in many fields, such as classification [7], objective detection [29], and semantic segmentation [21]. Some works introduce transformer blocks into the gait recognition framework. TransGait [24] proposes

a set transformer model with a temporal aggregation operation for obtaining set-level spatio-temporal features. SwinGait [10] utilizes convolutional blocks to extract silhouette feature and feed it to swinformer [30, 31] blocks. However, the former cannot enhance the local temporal receptive field, while the latter only obtains a window-global temporal receptive field. Differently, our GLGait employs Global-Local Temporal Module (GLTM) to both maintain global-local temporal receptive fields.

## 3 Method

In this section, we first introduce the pipeline of GLGait. Then, we detail the vision encoder, Global-Local 3D (GL-3D) block, and center-augmented triplet loss (CTL), respectively. Finally, we explain the optimization.

### 3.1 Pipeline

The pipeline of the proposed GLGait is shown in Figure 2 (a). A simple 2D convolution operation first initializes the silhouette sequences. Then we utilize the vision encoder to obtain a preliminary representation. After that, GL-3D blocks are used to extract both spatial information and global-local temporal information. Temporal Max Pooling operation (TP) and Horizontal Pooling operation [11, 14] are employed to aggregate the features. Finally, a combined loss function consisting of center-augmented triplet loss and cross-entropy loss is used to supervise the learning process.

### 3.2 Vision Encoder

Since gait silhouette is binary, containing limited information [10], we use a vision encoder to encode a preliminary pedestrian representation. Specifically, we use the conventional Pseudo 3D (P3D) blocks [16, 36] as the components. P3D block uses two 2D convolutions in the spatial dimension and a 1D convolution in the temporal dimension. Using $\mathbf{x}_{in} \in \mathbb{R}^{C \times T \times H \times W}$ as input, where $C$ is the channels, $T$ is frame number, $H$ and $W$ are the silhouette height

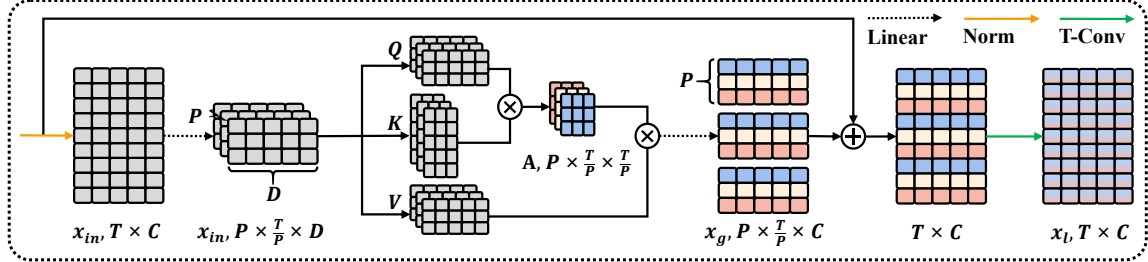

**Figure 3: Pseudo Global Temporal Self-Attention (PGTA) with a temporal convolution operation (T-Conv).**

and width, respectively. The process can be formulated as

$$\mathbf{x}_1 = \mathcal{R}(C_{2d}(\mathbf{x}_{in})), \tag{1}$$

$$\mathbf{x}_2 = C_{1d}(\mathbf{x}_1), \tag{2}$$

$$\mathbf{x}_3 = C_{2d}(\mathcal{R}(\mathbf{x}_1 + \mathbf{x}_2)), \tag{3}$$

$$\mathbf{x}_{out} = \mathcal{R}(\mathbf{x}_{in} + \mathbf{x}_3), \tag{4}$$

where $\mathbf{x}_{out} \in \mathbb{R}^{C \times T \times H \times W}$ is the output, $C_{1d}(\cdot)$ is the temporal convolution operation, $C_{2d}(\cdot)$ is the spatial convolution operation, $\mathcal{R}$ is the ReLU activation function.

### 3.3 GL-3D Block

**Operation.** As shown in Figure 2 (b), GL-3D block consists of two 2D convolutions for spatial information and a Global-Local Temporal Modul (GLTM) for the temporal feature. Specifically, GLTM mainly contains Pseudo Global Temporal Self-Attention (PGTA) and temporal convolution operation as shown in Figure 2 (c). Different from normal multi-head self-attention [44] (MHSA), PGTA places its focus on the temporal dimension, reducing the memory and computation complexity in two aspects. 1) PGTA separates the spatial dimension from the patch size, which means it is only 1D rather than 3D. 2) PGTA separates the patch size from the token inspired by [33]. We exhibit the structure of PGTA in Figure 3, where we only consider one head for simplicity.

Given $\mathbf{x}_{in} \in \mathbb{R}^{L \times T \times C}$ as input, where $L$ is $H \times W$, the formula of PGTA is as follows:

$$Reshaping \quad \mathbf{x}_{in}, \qquad \mathbb{R}^{L \times T \times C} \to \mathbb{R}^{L \times P \times \frac{T}{P} \times C} \tag{5}$$

$$[\mathbf{q}_i, \mathbf{k}_i, \mathbf{v}_i] = \mathbf{x}_{in}\mathbf{U}^i_{qkv}, \qquad \mathbf{U}^i_{qkv} \in \mathbb{R}^{C \times 3D} \tag{6}$$

$$\mathbf{A}_i = softmax(\mathbf{q}_i\mathbf{k}_i^T/\sqrt{D}), \qquad \mathbf{A}_i \in \mathbb{R}^{L \times P \times \frac{T}{P} \times \frac{T}{P}} \tag{7}$$

$$\mathbf{x}_i = \mathbf{A}_i\mathbf{v}_i, \qquad \mathbf{x}_i \in \mathbb{R}^{L \times P \times \frac{T}{P} \times D} \tag{8}$$

$$\mathbf{x}_g = [\mathbf{x}_1; \mathbf{x}_2; ...; \mathbf{x}_k]\mathbf{U}_{msa}, \qquad \mathbf{U}_{msa} \in \mathbb{R}^{k \cdot D \times C} \tag{9}$$

$$Reshaping \quad \mathbf{x}_g, \qquad \mathbb{R}^{L \times P \times \frac{T}{P} \times C} \to \mathbb{R}^{L \times T \times C} \tag{10}$$

where $i \in \{1, 2, ..., k\}$, $k$ is the number of multi-heads in PGTA, $P$ is patch size, $\mathbf{A}$ is attention matrix, $\mathbf{U}_{qkv}$ and $\mathbf{U}_{msa}$ are parameter matrices in PGTA. After PGTA, a temporal convolution operation is used to enhance the local temporal receptive field. Inspired by [15], we also add skip connections and 1D convolution operation with the kernel size of one as linear layer after PGTA to obtain a sub-update for each element in tokens. The process is formulated as

$$\mathbf{x}_l = \mathcal{R}(C_{1d}(\mathbf{x}_g + \mathbf{x}_{in})), \qquad \mathbf{x}_l \in \mathbb{R}^{L \times T \times D_{mid}} \tag{11}$$

$$\mathbf{x}_{out} = \mathbf{x}_l\mathbf{U}_{mlp} + \mathbf{x}_g, \qquad \mathbf{U}_{mlp} \in \mathbb{R}^{D_{mid} \times C} \tag{12}$$

where $\mathbf{x}_{out} \in \mathbb{R}^{L \times T \times C}$ is the output of GLTM, $\mathcal{R}$ and $C_{1d}(\cdot)$ are same as operations in Equation (1) and Equation (2), $D_{mid}$ is the channels of hidden layer, $\mathbf{U}_{mlp}$ is parameter matrix.

**Discussion.** In this subsection, we undertake a memory and computation complexity analysis of PGTA compared with a normal Spatio-Temporal MHSA [2, 44]. Specifically, we consider two kinds of $D$ in Equation (6). One is that $D$ is the same as the token size. Another is that $D$ is a scalar less than $C$. The input is $\mathbf{x}_{in} \in \mathbb{R}^{L \times T \times C}$. Given token size $P \times C$, we observe that the token size does not affect the computation complexity in Equation (6) and Equation (9) as follows,

$$C_{com} = O(\frac{LT}{P}PCD) \tag{13}$$

$$= O(LTCD). \tag{14}$$

Therefore, we only consider the computation complexity in Equation (7) and Equation (8). The formula of complexity in MHSA is as

$$C^M_{mem} = O(PCD), \tag{15}$$

$$C^M_{com} = O(N^2D), \tag{16}$$

where $N$ is $\frac{LT}{P}$, $C^M_{mem}$ is the memory complexity in MHSA, and $C^M_{com}$ is the computation complexity in MHSA. The formula of complexity in PGTA is as

$$C^P_{mem} = O(CD), \tag{17}$$

$$C^P_{com} = O(LP\frac{T^2}{P^2}D) \tag{18}$$

$$= O(LTND), \tag{19}$$

where $N$ is $\frac{T}{P}$, $C^P_{mem}$ is the memory complexity in PGTA, and $C^P_{com}$ is the computation complexity in PGTA.

First, we assume that $D$ is the same as the token size, which means information loss does not occur. For MHSA, given the token size $P_l \times P_t \times C$, where $P_l$ is spatial dimension size and $P_t$ is temporal dimension size, the memory complexity is $O(P_l^2 P_t^2 C^2)$ and the computation complexity is $O(\frac{L^2 T^2}{P_l P_t}C)$. For PGTA, given the token size $P_t \times C$, the memory complexity is $O(P_t C^2)$ and the computation complexity is $O(LT^2C)$. Compared with MHSA, PGTA reduces the memory complexity significantly, from $O(P_l^2 P_t^2 C^2)$ to $O(P_t C^2)$. For computation complexity, the decrease lies in $L$ and $P_l \times P_t$. Specifically, we set $P_t$ as 3 and $P_l$ as 4 in our experiments. Given $L$ as $16 \times 12$ in GL-3D block, $L$ is 16 times than $P_l \times P_t$, meaning PGTA reduces the computation complexity to 1/16 of MHSA.

Secondly, we assume that $D$ is a scalar less than $C$, which means information loss occurs, especially when the token size is large.

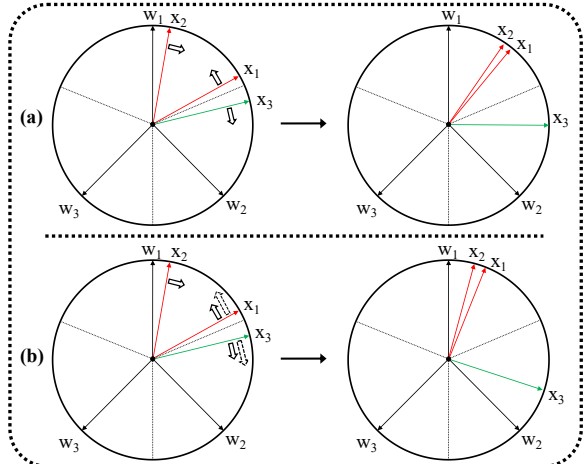

**Figure 4: Comparison of triplet loss [18] (a) and proposed center-augmented triplet loss (b). $w_i$ is class center in BN-Neck [32], $x_i$ is sample feature, dashed lines in the circle are class boundaries, $\Rightarrow$ is the gradient of the feature.**

For MHSA, given the token size $P_l \times P_t \times C$, the memory complexity is $O(P_l P_t CD)$ and the computation complexity is $O(\frac{L^2 T^2}{P_l^2 P_t^2} D)$. For PGTA, given the token size $P_t \times C$, the memory complexity is $O(CD)$ and the computation complexity is $O(L \frac{T^2}{P_t} D)$. PGTA also reduces the memory complexity significantly, from $O(P_l P_t CD)$ to $O(CD)$. For computation complexity, the decrease lies in $L$ and $P_l^2 \times P_t$. Following the same set above, PGTA reduces the computation complexity to a quarter of MHSA. Furthermore, MHSA loses much information. Assuming the feature dimension is a measure, the information loss of PGTA is only $O(C - D)$. For MHSA, the information loss even reaches $O(P_l P_t C - D)$.

Finally, considering the memory complexity, we set $D$ as a scalar in all our experiments.

### 3.4 Center-Augmented Triplet Loss

**Operation.** In contrast to the conventional triplet loss [18], the center-augmented triplet loss (CTL) additionally incorporates class centers as positive instances for each sample, the formulation is given as

$$\mathcal{L}_{ctl}(\mathbf{x}) = \sum_{i=1}^{Q+1} \sum_{j=1}^{N} \max(\mathcal{D}(\mathbf{x}, \mathbf{x}_i^q) - \mathcal{D}(\mathbf{x}, \mathbf{x}_j^n) + m, 0), \quad (20)$$

$$\mathcal{L}_{ctl} = \frac{1}{B} \sum_{k=1}^{B} \mathcal{L}_{ctl}(\mathbf{x}_k), \quad (21)$$

where $\mathbf{x} \in \mathbb{R}^C$ is input, $\mathbf{x}_i^q$ is the $i$-th positive sample of $\mathbf{x}$, $\mathbf{x}_j^n$ is the $j$-th negative sample of $\mathbf{x}$, $\mathcal{D}(\cdot, \cdot)$ is distance function, $m$ is margin, $Q$ is the number of positive samples, $N$ is the number of negative samples, $B$ is the batch size ($B = Q + N + 1$). Specifically, $\mathbf{x}_{Q+1}^q$ is the class center of $\mathbf{x}$, and Euclidean distance is used in $\mathcal{D}(\cdot, \cdot)$.

**Discussion.** In this subsection, we discuss the mechanism of center-augmented triplet loss (CTL). Apart from the common effects of triplet loss [18], CTL has two advantages.

First, CTL reduces intra-class distance. As illustrated in Figure 4 (a), for sample $\mathbf{x}_1$, $\mathbf{x}_2$ is a positive sample, and $\mathbf{x}_3$ is a negative one. When the distance between $(\mathbf{x}_1, \mathbf{x}_2)$ exceeds that of $(\mathbf{x}_1, \mathbf{x}_3)$, the triplet loss computes the corresponding gradient represented as solid-line arrows, encouraging $(\mathbf{x}_1, \mathbf{x}_2)$ to move closer, and $(\mathbf{x}_1, \mathbf{x}_3)$ to move apart. However, as $\mathbf{x}_2$ moves closer to $\mathbf{x}_1$, it keeps away the class center $\mathbf{w}_1$, thus increasing the intra-class distance. Conversely, as shown in Figure 4 (b), by treating $\mathbf{w}_1$ as a positive sample for $\mathbf{x}_1$ and $\mathbf{w}_3$ as one for $\mathbf{x}_3$, CTL calculates the respective gradients depicted as dashed-line arrows, driving $\mathbf{x}_1$ towards $\mathbf{w}_1$ and $\mathbf{x}_3$ towards $\mathbf{w}_3$. Due to the superposition of gradients from $(\mathbf{x}_1, \mathbf{x}_2)$ and $(\mathbf{x}_1, \mathbf{w}_1)$, $\mathbf{x}_1$ moves closer to $\mathbf{w}_1$, indirectly preventing $\mathbf{x}_2$ away from the class center $\mathbf{w}_1$, thus reducing the intra-class distance.

Secondly, CTL can directly increase the number of positive samples without expanding the batch size. In gait datasets such as Gait3D [52], many pedestrian silhouette sequences are limited in quantity, which means a lack of positive samples in a batch. CTL utilizes class centers as positive samples, adding computational complexity to the loss function without affecting the backbone computations of the model in the training stage and model inference in the test stage, presenting a good cost-effective trade-off.

### 3.5 Optimization

In the training stage of GLGait, a combined loss function consisting of center-augmented triplet loss ($\mathcal{L}_{ctl}$) and cross-entropy loss ($\mathcal{L}_{ce}$) is calculated to supervise the learning process,

$$\mathcal{L} = \alpha \mathcal{L}_{ctl} + \beta \mathcal{L}_{ce}, \quad (22)$$

where $\alpha$ and $\beta$ are hyper-parameters to balance the contributions to the total loss $\mathcal{L}$.

## 4 Experiments

In this section, we first introduce the datasets and implementation details. Then, we compare our proposed GLGait with the latest gait recognition methods and analyze the results. Finally, extensive ablation experiments prove the effectiveness of each component in GLGait.

### 4.1 Datasets and Implementation Details

The dataset information and implementation details in our experiments are as follows.

**Gait3D [52]** is a large scale gait dataset. Within a supermarket, 39 cameras capture 1,090 hours of videos with 1,920×1,080 resolution and 25 FPS. Through processing, a total of 4,000 subjects, 25,309 sequences, and 3,279,239 frame images are extracted. 3,000 subjects are compiled as the training set, while the remaining 1,000 subjects form the test set. For the testing phase, the probe comprises one sequence from each subject, and the gallery consists of the rest sequences.

**GREW [54]** is one of the largest gait datasets in the wild, including Silhouettes, GEIs, and 2D/3D human poses data types. The raw videos are collected from 882 cameras in large public areas. 7,533 video clips are used, containing nearly 3,500 hours of 1,920×1,080 streams. It has 26,345 subjects and 128,671 sequences, divided into two parts with 20,000 and 6,000 subjects as training set and test set,

**Table 1: Comparisons of Rank-1, 5 and 10 accuracies, mean Average Precision (%), and Parameter size (M) on Gait3D [52] and GREW [54] datasets.**

| Backbone Components | Method | Source | Gait3D | | | GREW | | | Params |
|---|---|---|---|---|---|---|---|---|---|
| | | | Rank-1 | Rank-5 | mAP | Rank-1 | Rank-5 | Rank-10 | |
| Convolution | GaitSet [4] | AAAI 2019 | 36.7 | 58.3 | 30.0 | 46.3 | 63.6 | 70.3 | 2.56 |
| | GaitPart [12] | CVPR 2020 | 28.2 | 47.6 | 21.6 | 44.0 | 60.7 | 67.3 | 1.46 |
| | GaitGL [28] | ICCV 2021 | 29.7 | 48.5 | 22.3 | 47.3 | 63.6 | 69.3 | 2.49 |
| | SMPLGait [52] | CVPR 2022 | 42.9 | 63.9 | 35.2 | - | - | - | - |
| | DyGait [46] | ICCV 2023 | 66.3 | 80.8 | 56.4 | 71.4 | 83.2 | 86.8 | - |
| | HSTL [45] | ICCV 2023 | 61.3 | 76.3 | 55.5 | 62.7 | 76.6 | 81.3 | 4.05 |
| | GaitGCI [9] | CVPR 2023 | 50.3 | 68.5 | 39.5 | 68.5 | 80.8 | 84.9 | - |
| | GaitBase [11] | CVPR 2023 | 64.3 | 79.6 | 55.5 | 59.1 | 74.5 | 78.9 | 4.90 |
| Convolution | DGaitV2-2D-B [10] | Arxiv 2023 | 64.5 | 81.7 | 56.5 | 62.3 | 76.4 | 81.5 | 2.35 |
| | DGaitV2-2D-L [10] | | 67.8 | 83.9 | 59.7 | 69.7 | 82.4 | 86.7 | 9.33 |
| | DGaitV2-P3D-B [10] | | 70.8 | 85.7 | 62.9 | 72.6 | 84.5 | 87.9 | 2.79 |
| | DGaitV2-P3D-L [10] | | 74.2 | 86.9 | 67.1 | 78.3 | 88.5 | 91.4 | 11.12 |
| | DGaitV2-P3D-H [10] | | 75.0 | - | - | 81.0 | - | - | 44.43 |
| | DGaitV2-3D-B [10] | | 71.0 | 85.0 | 62.3 | 73.1 | 84.9 | 88.4 | 6.92 |
| | DGaitV2-3D-L [10] | | 74.1 | 87.0 | 66.5 | 79.0 | 88.9 | 91.6 | 27.62 |
| | DGaitV2-3D-H [10] | | 75.8 | - | - | 81.6 | - | - | 110.44 |
| Convolution + Transformer | SwinGait-3D [10] | Arxiv 2023 | 75.0 | 86.7 | 67.2 | 79.3 | 88.9 | 91.8 | 13.1 |
| | GLGait-B | Ours | 73.9 | 86.3 | 65.9 | 75.4 | 86.3 | 89.6 | 3.58 |
| | GLGait-L | | 77.6 | 88.4 | 69.6 | 80.0 | 89.4 | 92.2 | 14.28 |
| | GLGait-H | | **77.7** | **88.9** | **70.6** | **82.8** | **91.1** | **93.5** | 57.04 |

respectively. Each subject in the test set has four sequences, two for probe and two for gallery.

**Implementation Details.** Our experiments are implemented using PyTorch [34]. We design the network capacity referring to the baselines [10, 11] as shown in Table 2. In Equation 22, $\alpha$ and $\beta$ are both set to 1. The kernel size of all convolution operations is 3. In Equation 8, $P$ is set to 3, and $D$ is the same as $C$ in Stage-1. In Equation 11, $D_{mid}$ is the same as $C$ in all stages. For parameters, we only consider the backbone without FC [4] and BNN [32] layers for all experiments. Similar to DGaitV2 [10], we also partition the model size into three segments: GLGait-Base (GLGait-B), GLGait-Large (GLGait-L), and GLGait-Huge (GLGait-H) to exhibit an appropriate compromise between accuracy and cost. They share identical architectures, except for the variation in the number of channels at each stage, which are (32, 64, 128, 256), (64, 128, 256, 512), and (128, 256, 512, 1024), respectively. During training, the input size of silhouettes is 64×44, and silhouettes are ordered in a sequence with a length of 30. The optimizer is the Stochastic Gradient Descent (SGD). The weight decay is 0.0005 and the momentum is 0.9. We adjust the learning rate, batch size, and the number of iterations to fit different dataset scales. 1) On Gait3D, we train the model for 120k iterations with a batch size of 32 × 4 (32 pedestrians, each containing 4 sequences). The learning rate starts at 0.1 and is subsequently decreased by a factor of 0.1 at iterations (40k, 80k, 100k). 2) On GREW, the model is trained for 180k iterations with a batch size of 32 × 4. The learning rate starts at 0.05 and is subsequently decreased by a factor of 0.2 at iterations (60k, 120k, 150k).

Specifically, we train the models in GaitBase [11] and DGaitV2 [10] by ourselves. As shown in Table 1, the results on the left side of the parentheses represent our results, while the results inside the parentheses originate from the original paper [10, 11].

## 4.2 Performance Comparison

We compare our approach with other recent appearance-based methods using silhouette sequence as input on in-the-wild datasets as shown in Table 1. Specifically, the input to SMPLGait [52] is exclusively comprised of the silhouette. These methods can be divided into two categories based on the network backbone components: one category backbone predominantly consists of convolutional operations, while another category backbone is constituted of both convolutional operations and transformers.

For the first category, the receptive field plays an essential role. For the spatial receptive field, as illustrated in Table 1, despite DGaitV2-2D-B [10] having half the parameters of GaitBase [11], its spatial receptive field substantially exceeds that of GaitBase, ultimately surpassing GaitBase in accuracy on Gait3D [52] and GREW [54] by 0.2% and 3.2%, respectively. The second consideration is the temporal receptive field. Under equivalent spatial receptive field conditions, DGaitV2-P3D-B outperforms DGaitV2-2D-B by 6.3% and 10.3% Rank-1 accuracy on Gait3D and GREW with only 0.44 MegaBytes parameters increase; a similar trend is observed with DGaitV2-P3D-L and DGaitV2-2D-L. However, no matter for DGaitV2-P3D or DGaitV2-3D, in comparison to sequences extending hundreds of silhouettes, their temporal receptive fields are significantly insufficient, merely extracting limited local temporal information.

For the second category, we consider that a global-local temporal receptive field shall be important. As depicted in Figure 1 (a), gait exhibits a certain cyclical pattern, and this cycle represents a local silhouette sequence within gait sequences. SwinGait-3D [10] utilizes a 3D residual block to encode a preliminary pedestrian representation, which is then fed into a 3D Swinformer [30, 31]

**Table 2: Network backbone of the GLGait.**

| Layer Name | Output Size | Structure $[k \times k \times k, c] \times b$ |
|---|---|---|
| Conv2D | $(T, C, 64, 44)$ | $[1 \times 3 \times 3, C] \times 1$ |
| Stage-1 Vision Encoder | $(T, C, 64, 44)$ | $\begin{bmatrix} 1 \times 3 \times 3, C \\ 3 \times 1 \times 1, C \\ 1 \times 3 \times 3, C \end{bmatrix} \times 1$ |
| Stage-2 Vision Encoder | $(T, 2C, 32, 22)$ | $\begin{bmatrix} 1 \times 3 \times 3, 2C \\ 3 \times 1 \times 1, 2C \\ 1 \times 3 \times 3, 2C \end{bmatrix} \times 4$ |
| Stage-3 GL-3D Block | $(T, 4C, 16, 11)$ | $\begin{bmatrix} 1 \times 3 \times 3, 4C \\ GLTM, 4C \\ 1 \times 3 \times 3, 4C \end{bmatrix} \times 4$ |
| Stage-4 GL-3D Block | $(T, 8C, 16, 11)$ | $\begin{bmatrix} 1 \times 3 \times 3, 8C \\ GLTM, 8C \\ 1 \times 3 \times 3, 8C \end{bmatrix} \times 1$ |

**Table 3: Extended experiment of sequence length on Gait3D with Rank-1 accuracy (%).**

| Method | Sequence Length | | | | | |
|---|---|---|---|---|---|---|
| | 1-100 | 101-200 | 201-300 | 301-400 | 401-500 | 1-500 |
| DGaitV2-P3D-L | 68.8 | 84.1 | 75.3 | 83.0 | 75.0 | 74.2 |
| GLGait-L | 71.7 | 84.1 | 76.4 | 88.1 | 85.7 | 76.6 |

**Table 4: Performance gain from applying CTL on Gait3D [52] and GREW [54] with Rank-1 accuracy (%).**

| Method | Gait3D | GREW |
|---|---|---|
| DGaitV2-P3D-B [10] | 70.8 → 72.1 | 72.6 → 74.3 |
| DGaitV2-P3D-L [10] | 74.2 → 75.4 | 78.3 → 79.6 |
| GLGait-B | 73.3 → 73.9 | 74.2 → 75.4 |
| GLGait-L | 76.6 → 77.6 | 79.7 → 80.0 |

**Table 5: Compared center-augmented triplet loss (CTL) with other loss function on Gait3D [52] with Rank-1 accuracy (%), where TL is triplet loss [18], CT is center loss [48], TCL is triplet center loss [17].**

| Method | TL | CL | TCL | CTL | Rank-1 |
|---|---|---|---|---|---|
| GLGait-B | √ | - | - | - | 73.3 |
| | - | √ | - | - | 72.8 |
| | - | - | √ | - | 73.3 |
| | - | - | - | √ | 73.9 |
| GLGait-L | √ | - | - | - | 76.6 |
| | - | √ | - | - | 76.1 |
| | - | - | √ | - | 76.2 |
| | - | - | - | √ | 77.6 |

block, achieving a window-global temporal receptive field. However, within these blocks, SwinGait-3D cannot obtain a true global one. In contrast, GLGait utilizes GLTM to exhibit a global-local temporal receptive field, thereby more effectively learning the cyclical motions of gait. With similar parameter counts, GLGait surpasses SwinGait-3D in Rank-1 accuracy both on Gait3D and GREW by 1.6% and 0.4%.

Moreover, we conduct an extended experiment to further explore the performance of GLGait across varying sequence lengths. The results are shown in Table 3, where the length distribution is illustrated in Figure 1 (b). GLGait-L outperforms DGaitV2-P3D-L 5.1% and 10.7% Rank-1 accuracy at lengths 301 to 400 and 401 to 500, respectively. This demonstrates that GLGait is effective in long sequences. Besides, we also observe that GLGait improves 2.9% Rank-1 accuracy at lengths 1 to 100, indicating that GLGait is even effective in short sequences rather than only in long sequences.

Finally, with the incorporation of CTL, GLGait-H achieves state-of-the-art performance on both Gait3D and GREW, obtaining Rank-1 accuracy of 77.7% and 82.8%, respectively.

**Effectiveness of Center-Augmented Triplet Loss.** To verify the effectiveness of CTL, we conduct ablation experiments on DGaitV2-P3D [10] and GLGait. As shown in Table 4, CTL improves both DGaitV2-P3D and GLGait compared with conventional triplet loss [18] on Gait3D [52] and GREW [54], demonstrating its generalizability and effectiveness. Meanwhile, we also compare CTL with center loss [48] (CL) and triplet center loss [17] (TCL) as shown in Table 5. CTL outperforms them in GLGait-B and GLGait-L. The reason lies in that CL and TCL only focus on the connection between samples and class centers, ignoring the pair of samples to samples. In contrast, CTL considers the pair of both samples to samples and samples to class centers, reducing intra-class distance

and expanding positive samples. CTL can seamlessly substitute conventional triplet loss [18], and we substitute it with CTL in subsequent experiments.

### 4.3 Ablation Experiments

We exhibit ablation experiments in GLGait to prove the effectiveness of each component.

**Vision Encoder Size.** To explore an appropriate vision encoder size, we conduct ablation studies within a controlled network, where the number of channels and blocks in per stage are fixed. Specifically, we employed the P3D block [16, 36] as the component of the vision encoder. As shown in Table 6, the model demonstrates optimal performance when S-1 and S-2 are both employed as the vision encoder, at which point the vision encoder is capable of learning an effective preliminary representation of pedestrians. Utilizing only S-1 as the vision encoder fails to obtain a satisfactory preliminary pedestrian representation, diminishing the model's learning efficiency within the GL-3D block. Conversely, incorporating S-1, S-2, and S-3 as the vision encoder does not afford additional space for the GL-3D block, impeding the model's ability to learn an effective global temporal receptive field and consequently degrading model performance. Finally, we employ S-1 and S-2 as the vision encoder.

**Vision Encoder Component.** We also exhibit the component ablation experiments on the vision encoder in Table 7. When employing P3D block [16, 36] as the component, GLGait obtains a better result with fewer parameters compared with 3D block [16, 36]. The possible reason lies in that our GL-3D block also separates the spatial and temporal dimensions, which is similar to P3D block. Maintaining such a similar structure assists in model training. Meanwhile, for 2D block [16], although it has fewer parameters, it is unable to process temporal information, which is essential in pedestrian representation, thus the performance significantly drops out.

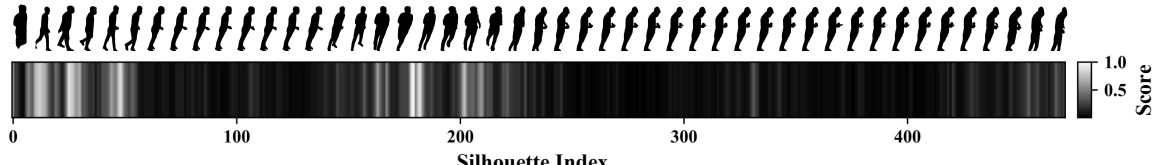

Figure 5: Silhouette score in Temporal Max Pooling phase, where the sequence contains 474 silhouettes from Gait3D.

Table 6: Ablation study of vision encoder size on Gait3D [52] with Rank-1 accuracy (%) and Params (M), where S-i is Stage-i, and checkmark (√) indicates that the stage is utilized as a part of the vision encoder.

| Method | S-1 | S-2 | S-3 | S-4 | Rank-1 | Params |
|---|---|---|---|---|---|---|
| GLGait-B | √ | - | - | - | 73.5 | 3.76 |
|  | √ | √ | - | - | 73.9 | 3.58 |
|  | √ | √ | √ | - | 72.6 | 3.12 |
| GLGait-L | √ | - | - | - | 76.6 | 15.00 |
|  | √ | √ | - | - | 77.6 | 14.28 |
|  | √ | √ | √ | - | 76.4 | 12.44 |

Table 7: Ablation study of vision encoder components on Gait3D [52] with Rank-1 accuracy (%), and Params (M).

| Method | Components | Rank-1 | Params |
|---|---|---|---|
| GLGait-B | 2D block [16] | 72.4 | 3.52 |
|  | 3D block [16, 36] | 73.1 | 4.12 |
|  | P3D block [16, 36] | 73.9 | 3.58 |
| GLGait-L | 2D block [16] | 75.2 | 14.07 |
|  | 3D block [16, 36] | 76.4 | 16.43 |
|  | P3D block [16, 36] | 77.6 | 14.28 |

Finally, we select P3D block as the component in the vision encoder to obtain a good accuracy and cost trade-off.

**Effectiveness of PGTA.** To verify the effectiveness of Pseudo Global Temporal Self-Attention (PGTA), we compare it with other multi-head self-attention [44] methods, containing Spatio-Temporal MHSA [2], Factorised self-attention [2] on temporal dimension, and MobileViT [33] self-attention. Specifically, we set patch size to $3 \times 4$. The results are shown in Table 8. PGTA reduces half of the parameters compared with Spatio-Temporal MHSA and Factorised self-attention. Meanwhile, we observe that the Rank-1 accuracy of Spatio-Temporal MHSA and Factorised self-attention greatly drops out. The possible reason is that a large information loss occurs between the 3,072 token size (patch size×channels) and 256 channels. Compared with MobileViT self-attention, PGTA improves 1.9% Rank-1 accuracy with fewer FLOPs. Due to the issue of the receptive field lying in the temporal dimension, PGTA only focuses on the temporal dimension and separates the spatial dimension from tokens, thus effectively establishing a good solution.

**Effectiveness of Temporal Convolution after PGTA.** To verify the effectiveness of temporal convolution after PGTA, we compare it with a normal linear operation. As shown in Table 9, employing temporal convolution improves GLGait-B 1.6% and GLGait-L 1.1% Rank-1 accuracy than a normal linear operation with few parameters increase. Temporal convolution enhances the local receptive field, assisting the model in learning the motion process of gait. Besides, temporal convolution can also aggregate pseudo

Table 8: Ablation study of memory and computation complexity in self-attention [44] on Gait3D [52] with Rank-1 accuracy (%), Params (M), and FLOPs (G).

| Method | Module | R-1 | Params | FLOPs |
|---|---|---|---|---|
| GLGait-B | Spatio-Temporal MHSA [2] | 68.2 | 7.93 | 0.93 |
|  | Factorised self-attention [2] | 70.6 | 7.93 | 0.92 |
|  | MobileViT self-attention [33] | 72.0 | 3.58 | 0.94 |
|  | PGTA | 73.9 | 3.58 | 0.87 |

Table 9: Ablation study of temporal convolution after PGTA on Gait3D [52] with Rank-1 accuracy (%), and Params (M).

| Method | Temporal Convolution | Rank-1 | Params |
|---|---|---|---|
| GLGait-B | - | 72.3 | 3.32 |
|  | √ | 73.9 | 3.58 |
| GLGait-L | - | 76.5 | 13.23 |
|  | √ | 77.6 | 14.28 |

global temporal receptive fields generated by PGTA to a true holistic temporal receptive field. Its effectiveness is well demonstrated.

### 4.4 Visualization

To verify the effectiveness of GLGait in long sequences, we conduct visualization as illustrated in Figure 5, where the score is model attention in temporal max pooling phase for each silhouette. GLGait can detect dynamic sub-sequences and give them high scores; for static sub-sequences, it selects representative silhouettes to give high scores and assigns low scores to the rest. This demonstrates that GLGait can align various gait patterns in long sequences, thus validating the effectiveness of global-local temporal receptive field.

### 5 Conclusion

In this paper, to address the issue of temporal receptive field for gait recognition in the wild, we add the multi-head self-attention (MHSA) before temporal convolution operation in Convolutional Neural Networks (ConvNets), designing a Global-Local Temporal Receptive Field Network (GLGait) to obtain a global-local temporal receptive field. Due to the dimension explosion in MHSA, we propose a Pseudo Global Temporal Self-Attention (PGTA) to reduce the memory and computation complexity. Furthermore, a Center-Augmented Triplet Loss (CTL) is proposed to reduce the intra-class distance and expand the positive samples, seamlessly substituting conventional triplet loss. GLGait can effectively recognize pedestrians with limited memory and computation complexity in wild scenarios, thus this work can be applied to widespread surveillance gait recognition systems. Extensive experiments have been conducted on Gait3D and GREW. The results demonstrate that our approach outperforms state-of-the-art methods on in-the-wild datasets.

## Acknowledgments

This work was supported by the Key Program of National Natural Science Foundation of China (Grant No. U20B2069) and research funding from Kuaishou Technology.

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
