# OpenReview forum: "GLGait: A Global-Local Temporal Receptive Field Network for Gait Recognition in the Wild"
_acmmm.org/ACMMM/2024/Conference — MM2024 Poster_

### Official Review · Reviewer_f5QR · 2024-05-14

**Rating:** 4
**Confidence:** 4

**Summary:**

The paper is focused on the task of gait recognition in the wild, which aims to identify pedestrians by analyzing their walking patterns. The main idea of this paper is to solve the limited temporal receptive field of existing appearance-based methods for gait recognition in the wild. Therefore, this paper makes the following contributions: 1) a Global-Local Temporal Receptive Field Network (GLGait) for gait recognition in the wild, 2) the Pseudo Global Temporal Self-Attention (PGTA) to reduce the memory and computation complexity of multi-head self-attention (MHSA), and 3) a Center-Augmented Triplet Loss (CTL) for training GLGait. Experiments on two public datasets show the effectiveness of the proposed method.

**Strengths:**

1. Novelty: This paper proposes a Global-Local Temporal Receptive Field Network with the Pseudo Global Temporal Self-Attention modules. This approach is novel in the community of gait recognition.
2. Technical correctness: The proposed Pseudo Global Temporal Self-Attention and Center-Augmented Triplet Loss are technically correct.
3. Clarity: The proposed GLGait network is presented in a clear and concise manner, with a detailed explanation of the components such as the Pseudo Global Temporal Self-Attention and Center-Augmented Triplet Loss.
4. Applications: The proposed GLGait network has potential applications in various gait-related domains such as surveillance and assistive technology for elderly people.

Overall, this paper presents a novel approach for gait recognition in the wild, which is based on a clear and concise approach and has been demonstrated to be effective through experiments on two in-the-wild datasets. The paper also provides a comprehensive comparison with other state-of-the-art methods for gait recognition in the wild, making it a valuable contribution to the field.

**Limitations:**

1. Basically, this paper considers unimodal data in the gait recognition task and is less suitable for the conference.
2. The technical novelty of the Center-Augmented Triplet Loss is incremental to the existing triplet-like loss.
3. The experimental results are somewhat insufficient. This paper only provides quantitative results on the two datasets but no qualitative or visualized results to intuitively validate the claim of this paper, i.e., the proposed method can solve the problem of long-term temporal gait sequence modeling. I suggest the authors provide some examples as shown in Figure 1 to reflect that the propose method can capture the important silhouettes in a long sequence.

**Suitability:**

2

---

### Official Review · Reviewer_wV1a · 2024-05-15

**Rating:** 3
**Confidence:** 4

**Summary:**

This paper is dedicated to the challenge of gait recognition in the wild. The authors propose the Global-Local Temporal Receptive Field Network (GLGait) to obtain more robust temporal information from gait sequences. The PGTA module is developed to alleviate computational complexity and reduce memory requirements. The experimental results show the effectiveness of the proposed method.

**Strengths:**

1)	The PGTA module is designed to reduce the high memory cost and computation complexity of multi-head self-attention.
2)	The conducted experiments are comprehensive and demonstrate the effectiveness of the proposed method.

**Limitations:**

1)	The title of the paper is about global-local temporal receptive field Network, but the method section lacks relevant descriptions of this content.
2)	The complexity analysis in the method section is too much, so it is suggested to trim it appropriately, and the related content can be moved to supplementary materials.
3)	In Figure 2 (a), both P3D and GL-3D are depicted twice, please note the difference in the figure and caption.
4)	In Figure 2, please indicate the meaning of the parameter p, and indicate whether the value of p is exactly the same in different places.
5)	Lines 327 and 336, C, H, and W should be distinguished by different symbols, such as lowercase letters.
6)	There is a lack of correspondence between Formulas 5-10 and Figure 3, which hampers reader comprehension. It would be beneficial to clearly mark corresponding variables such as $X_{in}$, $X_{i}$, etc., in Figure 3.

**Suitability:**

2

---

### Official Review · Reviewer_6joL · 2024-05-21

**Rating:** 4
**Confidence:** 4

**Summary:**

The paper introduces a Global-Local Temporal Receptive Field Network named GLGait, designed to tackle the issue of temporal receptive fields in gait recognition in-the-wilds. By integrating Pseudo Global Temporal Self-Attention (PGTA) with temporal convolution operations, GLGait effectively processes long gait sequences while reducing the model’s memory and computational complexity. Additionally, the paper proposes a novel loss function, the Center-Augmented Triplet Loss (CTL), to decrease intra-class distances and increase positive samples during the training phase. Extensive experiments demonstrate that GLGait achieves state-of-the-art performance on in-the-wild datasets.

**Strengths:**

a. This paper achieved state-of-the-art performance on the in-the-wild datasets Gait3D and GREW.
b. This paper demonstrates how to reduce the model’s complexity while maintaining technical accuracy by introducing PGTA and CTL.
c.This paper articulates the methodology, experimental design, and results analysis with clarity.

**Limitations:**

a. Although GLGait introduces the concept of a global-local temporal receptive field and PGTA, these ideas may overlap with existing research. The integration of global and local features has been extensively explored in the field of deep learning.
b. This paper may not provide sufficient details to fully understand the implementation of PGTA and CTL, nor does it mention any intention to publish the code.
c. Robust gait recognition methods need to maintain accuracy and stability over time, and the paper does not discuss the model’s performance over time spans.

**Suitability:**

3

---

### Official Review · Reviewer_35hV · 2024-05-25

**Rating:** 4
**Confidence:** 3

**Summary:**

This paper focuses on gait recognition applied in the wild instead of in the lab scenarios. According to the challenge of the wild scenarios (especially in the representation of temporal dimension), the authors propose a Global-Local Temporal Receptive Field Network (GLGait) to establish a global-local temporal receptive field. Furthermore, this paper also proposes a Center-Augmented Triplet Loss (CTL) in GLGait to reduce the intra-class distance and expand the positive samples in the training stage.

**Strengths:**

This paper proposes a novel Global-Local Temporal Receptive Field Network (GLGait) to obtain a global-local temporal receptive field for gait recognition in the wild. Besides, the paper proposes a Center-Augmented Triplet Loss (CTL) to assist in model training. Extensive experiments demonstrate the proposed approach obtains the state-of-the-art performance on in-the-wild datasets.

**Limitations:**

1. This paper mentioned that the previous ConvNets-like methods is significantly insufficient to obtain the temporal receptive field. As we know, some operator in of ConvNets also can enlarge the receptive field to some extent，such as “Dilated Convolution”. Please provide the reason that employ “transformer” instead of “Dilated Convolution”?
2. In the subsection 3.5, the hyper-parameters α and β in formula (22) are utilized to balance the contributions to the total loss. How to determine the two hyper-parameters?

**Suitability:**

2

---

### Meta-Review · Area_Chair_M2N2 · 2024-06-28

**Recommendation:** Accept (Poster)
**Confidence:** 5

**Metareview:**

All reviewers agree that this paper is well introduced, motivated and described, with enough level of novelty. Although reviewers provided some critiques/suggestions to improve clarity in some parts, all reviewers agree that the significance of the work and the provided evaluation are good for acceptance of the paper.